# Premature Atrial and Ventricular Contraction Detection Using Photoplethysmographic Data from a Smartwatch

**DOI:** 10.3390/s20195683

**Published:** 2020-10-05

**Authors:** Dong Han, Syed Khairul Bashar, Fahimeh Mohagheghian, Eric Ding, Cody Whitcomb, David D. McManus, Ki H. Chon

**Affiliations:** 1Department of Biomedical Engineering, University of Connecticut, Storrs, CT 06269, USA; dong.han@uconn.edu (D.H.); syed.bashar@uconn.edu (S.K.B.); fahimeh.mohagheghian@uconn.edu (F.M.); 2Division of Cardiology, University of Massachusetts Medical School, Worcester, MA 01655, USA; eric.ding@umassmed.edu (E.D.); cody.whitcomb@tufts.edu (C.W.); david.mcmanus@umassmed.edu (D.D.M.)

**Keywords:** premature atrial contraction detection, Poincaré plot, premature ventricular contraction detection

## Abstract

We developed an algorithm to detect premature atrial contraction (PAC) and premature ventricular contraction (PVC) using photoplethysmographic (PPG) data acquired from a smartwatch. Our PAC/PVC detection algorithm is composed of a sequence of algorithms that are combined to discriminate various arrhythmias. A novel vector resemblance method is used to enhance the PAC/PVC detection results of the Poincaré plot method. The new PAC/PVC detection algorithm with our automated motion and noise artifact detection approach yielded a sensitivity of 86% for atrial fibrillation (AF) subjects while the overall sensitivity was 67% when normal sinus rhythm (NSR) subjects were also included. The specificity, positive predictive value (PPV), negative predictive value (NPV) and accuracy values for the combined data consisting of both NSR and AF subjects were 97%, 81%, 94% and 92%, respectively, for PAC/PVC detection combined with our automated motion and noise artifact detection approach. Moreover, when AF detection was compared with and without PAC/PVC, the sensitivity and specificity increased from 94.55% to 98.18% and from 95.75% to 97.90%, respectively. For additional independent testing data, we used two datasets: a smartwatch PPG dataset that was collected in our ongoing clinical study, and a pulse oximetry PPG dataset from the Medical Information Mart for Intensive Care III database. The PAC/PVC classification results of the independent testing on these two other datasets are all above 92% for sensitivity, specificity, PPV, NPV, and accuracy. The proposed combined approach to detect PAC and PVC can ultimately lead to better accuracy in AF detection. This is one of the first studies involving detection of PAC and PVC using PPG recordings from a smartwatch. The proposed method can potentially be of clinical importance as this enhanced capability can lead to fewer false positive detections of AF, especially for those NSR subjects with frequent episodes of PAC/PVC.

## 1. Introduction

Premature atrial contractions (PAC) and premature ventricular contractions (PVC) are benign cardiac arrhythmias that occur in some healthy subjects; their impact on overall cardiovascular well-being is still unclear [1]. Although the debate is ongoing, there is no clear clinical evidence to suggest that either PAC or PVC are malignant arrhythmias or that their presence can trigger fatal arrhythmias. However, it is clear that frequent occurrence of PAC and PVC can lead to incorrect detection of atrial fibrillation (AF), as irregular ectopic beats interspersed among normal sinus rhythm (NSR) can lead to sufficient appearance of randomness, hence, false positive detection [2]. AF is the most commonly encountered arrhythmia in clinical practice and it has been forecasted that over 6 million Americans will be affected by 2050 [3]. As a significant percent of the population suffering from AF are asymptomatic, and most will likely have paroxysmal—short lasting, intermittent and random onset—AF (pAF), there has been a growing need for long term monitoring. Wearable devices such as Holter and event monitors, and recently cardiac patches, have been used to monitor pAF [4,5]. Most recently, there has been growing interest in using smartphones and especially smartwatches for detecting AF [2,3,6], as these devices are especially practical for long-term monitoring. While the form factor of smartwatches for long-term pAF monitoring is convenient and potentially widely accepted by users, the major drawback is that the photoplethysmogram (PPG) is measured using light emitting diodes and photodetectors are quite sensitive to motion artifacts. Moreover, the capability to detect other arrhythmias than just AF has not yet been developed for smartwatch-based PPG signals.

Algorithms for PAC and PVC detection using electrocardiographic (ECG) data have been well established, and due to their distinctive waveform morphologies, these rhythms can be accurately distinguished [7,8]. For example, Chetan et al. [8] proposed a method that can decompose ECG signals and extract non-linear features from the ECG waveform to discriminate ventricular arrhythmias from atrial arrhythmias. However, a PPG waveform associated with ventricular arrhythmia does not have a distinguishable T-wave in the ECG that is different from the ECG waveform of atrial arrhythmia. Thus, the method proposed by Chetan et al. [8] cannot be used for PPG signals. Fortunately, the Lorentz plot obtained from RR interval time series is implemented in [9] to detect AF and atrial tachycardia including PACs. Recently, low resolution Lorentz plot images were trained with a convolutional neural network to discriminate between AF and sinus rhythm [10]. In another recent study, RR intervals and QRS morphology-based features were combined with a random forest to detect PVC in [11]. However, there is scant literature on detecting PAC/PVC from PPG-based data. Recently, our lab has published about PAC and PVC detection using a Poincaré plot method applied to fingertip video-PPG data acquired from a smartphone [2]. It should be noted; however, that video-PPG signal quality from a fingertip is much higher than that from a wrist, as the skin is thinner on the fingertip. Moreover, motion artifacts are less severe for video PPG data from a smartphone than for smartwatch PPG data, as long as the finger makes constant good contact with the smartphone’s camera. A study by Solosenko et al. used an adaptive low-pass filtered PPG waveform of premature beats to train an artificial neural network to detect PVC, with good results [1]. They took fingertip PPG datasets from pulse oximeters out of the MIMIC and MIMIC II databases. However, their intent was not to discriminate between PVC and other types of cardiac arrhythmia such as AF. In [12], PPG pulse rate turbulence onset, slope and turbulence shape are used to classify PVCs from NSR; however, PAC and AF were not studied in [12]. In [13], PPG morphology based algorithm is proposed to reject non-sinus rhythm pulses; although no discrimination is made among different pulses. Signal quality assessment and suppression of ectopic beats, bigeminy along with respiratory sinus arrhythmia are described in [14] for more accurate AF detection; however, no detection of PACs or PVCs are reported. In [15], the case of ventricular arrhythmia (VA) was considered in AF detection from wrist PPG signal, however, only accuracy of AF detection was reported, and no independent VA detection was performed. Recently, deep learning method is used to classify AF and NSR in [16] which mentioned that high PAC burden would lower AF detection accuracy. However, deep neural network models proposed in this paper did not discriminate PACs from NSR and AF. Moreover, the dataset used in this paper did not consider PVC arrhythmia. Many other studies on AF detection using wrist PPG sensors were conducted [17,18,19,20,21,22,23,24,25,26,27,28,29,30,31,32], however, none of them considered PAC or PVC case in their AF detection.

To the best of our knowledge, this is one of the first studies which demonstrates detection of PAC and PVC from smartwatch PPG data. To discriminate amongst NSR, AF, PAC, and PVC, a combination of PAC and PVC detection algorithms is designed to distinguish PAC/PVC from AF and NSR using the heart rates derived from a peak detection method, as illustrated in Figure 1.

The evaluation of our proposed method was based on data collected from both NSR and AF subjects using Samsung Simband 2 smartwatches at the University of Massachusetts Medical Center (UMMC).

## 2. Materials and Methods

### 2.1. Datasets and Experimental Protocol

#### 2.1.1. Training Dataset: UMMC Simband Dataset

37 patients (28 male and 9 female) with cardiac arrhythmia, ranging in age from 50 to 91 years old, participated in the smartwatch study at the ambulatory cardiovascular clinic at University of Massachusetts Medical Center (UMMC). This population is representative for AF, PAC and PVC subjects as described in [33]. The details of subject characteristics, monitoring duration, and arrhythmia burden are listed in Table 1. The Institutional Review Boards at both UMMC and the University of Connecticut approved the study protocol. Reference ECG and smartwatch data were simultaneously measured from the chest and wrist using a 7-lead Holter monitor (Rozinn RZ153+ Series, Rozinn Electronics Inc., Glendale, NY, USA) and Simband 2 (Samsung Digital Health, San Jose, CA, USA) (henceforth referred to simply as Simband), respectively. ECG data consisted of 3-channel signals, each sampled at 180 Hz. The Holter ECG data were used as the reference. Simband data consisted of 8-channel PPG signals, three-axis accelerometers and a one-lead ECG which was obtained by touching the wristband’s stainless-steel electrode with a finger from the non-watch worn hand. Simband PPG signals were sampled at 128 Hz. Only the 5th PPG channel (green LED color, wave length 520–535 nm) was used for data analysis since it consistently provided the best quality signal. The subjects were requested to touch the Simband’s ECG electrode for at least one minute at the beginning of data collection so that PPG data could be aligned with Holter ECG data. The alignment of the Simband and Holter ECG signals was performed by estimating the cross-correlation between the signals [34]. When two signals are well-aligned, the cross-correlation value will have a value >0.8 and if they are not, it can be lower than 0.4. However, even for perfectly aligned signal, the cross-correlation value will be lower than 1 if they have a low signal-to-noise ratio. For this reason, we chose 0.9 for the alignment of ECG data from Simband and Holter monitors. We manually checked both ECG signal at the point where alignment could occur when cross-correlation is around 0.9, and we slightly adjusted the alignment time point to make sure R waves in both ECG signals were precisely aligned. Given this need for correlation, only the Holter ECG data with good signal quality (discernible R waves) were retained, as they also needed to be used as the reference heart rates from which PPG heart rate estimates were benchmarked. PPG and accelerometer data were down sampled to 50 Hz and 20 Hz, respectively. All signals were segmented into 30-s lengths, with no overlap, for peak detection analysis in real time clinical detection. This dataset was created as a part of a preliminary study, thus we have a limited number of subjects and recordings. The training dataset is available for download on our lab’s website listed in Appendix A.

Our experimental protocol was designed to simulate everyday activities and consisted of the following sequence: sit still for 2 min, walk slowly for 2 min, stand still for 30 s, walk quickly for 2 min, stand still for 30 s, make watch-arm movements for 2 min, and stand still for 30 s. Patients were then asked to sit and stand repetitively for 1 minute followed by climbing stairs for 1 additional minute. The last procedure required patients to sit still for 1 minute. Figure 2 succinctly summarizes the above-described protocol.

#### 2.1.2. Testing Dataset: Samsung Gear S3 Dataset

For the first independent testing dataset, we randomly selected two subjects with 101 30-s segments from patients who participated in the ongoing clinical trial (funded by NIH) at UMMC. The Institutional Review Boards at UMMC approved the study protocol. Subject 005 was non-AF with 54 NSR segments and 27 PAC/PVC segments. Subject 307 was an AF subject with 20 clean AF segments. The reference ECG and smartwatch data were simultaneously measured from the chest and wrist using a 2-lead rhythm patch device (Cardea SOLO, Cardiac Insight Inc., Bellevue, WA, USA) and Samsung Gear S3 (Samsung, San Jose, CA, USA) (henceforth referred to simply as Gear S3), respectively. (This is a different smartwatch than the Simband we used for the training data collection.) The patch ECG data consisted of 1-channel signals and were sampled at 250 Hz. The patch ECG data were used as the reference. The Gear S3 data consisted of 1-channel PPG signal and 1-channel magnitude of the accelerometer signal. Gear S3 signals were all sampled at 50 Hz and were automatically segmented into 30-s lengths. The alignment of the Gear S3 and ECG patch signals for every 30-s segment was done by visual inspection. The heart rates from the patch ECG and PPG signals were based on Hossain et al.’s method [35] and Han et al.’s method [36], respectively.

The enrolled patients wore the Gear S3 smartwatch and ECG patch 24 h a day with no restriction on their regular daily activities, for 14 consecutive days, to evaluate the performance of our real-time PAC/PVC detection algorithm. Due to the 7-day battery limitation, patients switched to a second new ECG patch on the 7th day of trial. Smartwatches were charged daily for 1 h.

#### 2.1.3. Testing Dataset: MIMIC III Dataset

The recently released Medical Information Mart for Intensive Care (MIMIC III) database provides continuous ECG and pulse oximetry waveforms (PLETH) from patients in critical care at a large tertiary care hospital [37,38]. Ten subjects, listed in Table 2, were selected from MIMIC III database [39].

Since the ICU recording for each subject in the MIMIC III [38] dataset contained hundreds of hours of data, we only used the subjects whose data had already been prepared for the AF study [39], in which cardiologists had spent several months to adjudicate the presence of AF in those recordings. The AF detection study described in [39] was based on long-term ECG recordings, where the subjects were selected from critically ill sepsis patients. Since the PPG signal was neither analyzed nor observed in that study, only a few subjects in that study had simultaneous PPG and ECG signals. However, for this paper we needed subjects with simultaneous ECG and PPG recordings, as well as AF annotations from the cardiologists. As a result, we manually selected a random subset of 10 subjects from the study population of [39] for whom simultaneous ECG and PPG signals were available, and used this subset of 10 subjects as a separate blind test dataset.

Approximately four hours of PPG and ECG data were used, with the former for testing. The ECG was used for PPG beats validation. All signals were originally sampled at 125 Hz. PLETHs signal were down-sampled to 50 Hz to be concordant with the training dataset. All signals were segmented to a 30-s length with no overlap for peak detection analysis. The testing dataset is also available on our lab’s website listed in Appendix A.

### 2.2. Methods and Evaluation

#### 2.2.1. Overview of PAC/PVC Detection

Figure 1 shows the diagram of our proposed PAC/PVC detection approach. Motion artifact detection [40] is performed as the first step to ensure that PPG data are relatively clean. The algorithm involves a combination of features derived from the time-frequency analysis of PPG and the significance of the accelerometer magnitude. To briefly summarize the motion artifact detection [40]: we used two signals, the accelerometer (ACC) and PPG signals, to decide if motion noise existed in a 30-s segment. Motion artifact detection for the ACC signal was based on a thresholding method on the mean value of the univariance ACC data. For detecting motion artifacts in the PPG signal, a time-frequency spectra (TFS) was first calculated from the univariance and zero-mean PPG, and then two signals’ power consisting of the clean ($Q_{1}$) and noisy ($Q_{2}$) signals, were calculated from the TFS. By counting the time instances of $Q_{1}$ and $Q_{2}$, we can separate the corrupted PPG signal from the clean PPG signal. We considered a 30-s PPG segment as clean only when both the PPG segment and its corresponding ACC segment were detected as clean. The details of our motion artifact detection algorithm have been described in [40].

If PPG data pass the motion artifact detection test, peak detection is performed. The details of our peak detection algorithm have been described in [36]. The approach defined as the overall sequence for waveform envelope peak detection is named SWEPD and it is shown to be accurate for various arrhythmias including AF [36]. The next step is to perform AF detection [40] to separate AF from NSR. PAC/PVC detection is performed on both AF and NSR data segments. A modified version of the PAC/PVC detection algorithm for NSR is applied to AF detected data via our algorithm to ensure fewer false positive detections of AF. Details of each of these steps are provided in the next sections.

#### 2.2.2. Poincaré Plot for PAC/PVC Pattern Recognition

The aim of this section is to expose distinct patterns of PAC/PVC heart rates that distinguish them from NSR and AF rhythms. The PAC/PVC detection method is similar in concept to the Poincaré plot algorithm developed for smartphone video camera data [2], but modifications were necessary to be applicable to smartwatch PPG data, which are more prone to motion and noise artifacts. The Poincaré plot was originally used for heart failure detection using ECG data [41]. The rationale behind using the Poincaré plot is that it allows assessment of the nonlinear dynamics of the RR interval data, which are usually non-Gaussian in distribution. It is simple to implement method and allows good visualization of the nonlinear dynamic characteristics of the data [41]. Figure 3 shows four representative examples of PPG signal 30-s segments (left panels) and the corresponding Poincaré plots (right panels) from subjects with four different rhythms: NSR, PVC, normal heart rate AF and fast heart rate AF. Normal and fast heart rate AF are defined as AF episodes with the mean heart rates below, and greater than or equal to, 100 beats/min, respectively. The Poincaré plot is based on the difference in heart rates of the current beat to the next beat, on the *X*-axis, and of the previous beat to the current beat, on the *Y*-axis, for the entire 30-s data segment in the left panels of Figure 3. In panel (2) of Figure 3, a distinct repeated kite shape pattern emerges for PAC/PVC rhythms. The difference in heart rates for the *X*- and *Y*-axes of the Poincaré plots are set to [−80, 80] beats per minute (BPM). Moreover, the Poincaré plot is subdivided into nine quadrants, which represent permutations of all possible sequences of “long” and “short” time differences based on three consecutive heart beat intervals. A “long” time difference occurs when a time difference of the heartbeats associated with either the PAC or PVC beat to the next normal heartbeat is compared to the time difference between two consecutive normal heartbeats. A “short” time difference occurs when a time difference between a previous normal beat to either a PAC or PVC beat is compared to the time difference between two consecutive normal heartbeats. The quadrants are divided to capture PAC/PVC patterns as well as to discriminate between normal and fast heart rates of AF and PAC/PVC. These quadrants’ boundaries were set optimally based on experimental data. The initial assumption for the width of the zeroth quadrant is set to 10 BPM, symmetrically starts from -5 and ends on 5 BPM on both X and Y axes based on the maximum variation of 10 BPM heart rate that sinus rhythm subject could have [42]. Variations of adding or subtracting 3 BPM from the 10 BPM width were tested to optimize the PAC/PVC detection, and the best width is still 10 BPM. As we finalized the range for zeroth quadrant, widths for the rest eight quadrants are easy to set. The widths of the shorter side for quadrant one, three, five and six are the same size of zeroth quadrant; their longer width is set to 75 BPM, which is equal to subtraction of the half width of Poincaré plot with the half width of zeroth quadrant. For quadrant two, four, seven and eight, their widths are the same as the widths of longer side for quadrant one, three, five and six.

All possible Poincaré plot quadrant patterns associated with three consecutive heartbeats are provided in Table 3. In addition, Table 4 provides various arrhythmias with their corresponding trajectory patterns in Poincaré plots. The right panels from the top to bottom of Figure 3 show typical patterns for NSR, PAC/PVC, normal heart rate AF and fast heart rate AF, respectively. As shown in these figures, there are distinct patterns so that discrimination among these rhythms can be made.

In our previous study, we showed only two types of PAC/PVC pattern, with the quadrant sequence ‘1-2-3’ for trigeminy PAC/PVC and ‘2-4’ for bigeminy PAC/PVC [2]. However, in this study, we include a new pattern that resides in the quadrants ‘6-4-5’ to represent fast heart rate PAC/PVC. These PAC/PVC patterns can be seen in the right panels of Figure 4. In Figure 4, the five rows of panels (labelled 1-5) show five different types of PAC/PVC patterns from the Poincaré plots. The top to bottom strips (labelled as a, b, c and d) shown within each of the left panels of Figure 4 show ECG, raw PPG, filtered PPG and comparison of the reference and PPG-derived heart rates, respectively, for representative data segments. In the plots (c) of the left panels, we also labeled Poincaré quadrants associated with each PPG peak. The right panels of Figure 4 are the corresponding Poincaré plots derived from the difference in heart rates obtained from the peak detection shown in the plots (b) of the left panels of Figure 4.

The top right panel of Figure 4 shows a normal heart rate PAC with the ‘1-2-3’ pattern, having the appearance of an upward kite on the Poincaré plot. Row 2’s right panel shows a fast heart rate PAC segment with the ‘6-4-5’ pattern, with the appearance of a downward kite on the Poincaré plot.

The third row’s right panel of Figure 4 shows a fast heart rate PVC segment with the ‘6-4-5’ downward kite shape pattern on the Poincaré plot.

The right panel of row 4 in Figure 4 shows the pattern of ‘2-4’ to reflect bigeminy and ‘6-4-5’ to signify fast heart beats in the Poincaré plots, whereas the fifth row’s right panel represents a slow heart rate PAC and largely shows ‘2-4’ or bigeminy pattern. We use the three heart rate patterns of ‘6-4-5,’ ‘1-2-3,’ and ‘2-4’ to detect PAC/PVC.

#### 2.2.3. Vector Resemblance of Poincaré Plot Trajectory for PAC/PVC Detection in AF-Detected Subjects

To automatically quantify the patterns associated with PAC/PVC rhythms described in the preceding section, we propose a method called vector resemblance to represent the similarity of PAC/PVC trajectories. Note that frequent occurrences of PAC/PVC can erroneously lead to false positive detection of AF [2], hence, it is important to discriminate them from AF rhythms. Before introducing our algorithm, we must note that the most important criterion in our paper is that either the ‘1-2-3’ or ‘6-4-5’ patterns need to repeat more than three times in a 30-s segment to be considered as PAC/PVC. The biggest reason for this choice is that our AF detection algorithm [40] will falsely detect AF when at least three heart beats in a 30-s segment are ectopic beats. Another reason is that PAC/PVC often occurs more than once, as evidenced by bigeminy and trigeminy occurrences of these rhythms. Moreover, three PAC/PVC patterns ensures better detection accuracy, as either random motion artifacts or AF can sometimes lead to the above-noted kite-shaped patterns associated with PAC/PVC.

Algorithm 1 summarizes the pseudo-code of our proposed PAC/PVC detection algorithm for those segments initially detected as AF.

As shown in Figure 1, AF detection is first performed to discriminate subjects with AF from NSR. For both AF and NSR subjects, PAC/PVC detection is subsequently performed. Hence, we code for automatic detection of PAC/PVC intervals in segments that also contain AF. The vectors are shown in the Poincaré plots of Figure 4, which show trajectories from the start to their destination. Since a PAC/PVC pattern can have the ‘6-4-5’ kite shaped pattern, angles for the vector ‘6-4’ and ‘4-5’ are calculated. The same procedure is used for the pattern ‘1-2-3’. The pattern ‘2-4’ is not considered for PAC/PVC detection since AF can mimic this pattern due to heart rates that can go from normal and fast. However, the pattern ‘2-4’ is used for PAC/PVC detection in NSR as described in the section pertaining to non-AF subjects. We use a threshold of the standard deviation of the vector angles to sort out similar vectors so that those PAC/PVC occurrences which lead to false positive AF detection can be reclassified as PAC/PVC.

The first threshold involves the standard deviation of vector angles. Standard deviation is a measure of the trajectory resemblance on Poincaré plots. Because PAC/PVC occurrences are repeated (e.g., bigeminy, trigemini, etc.) the trajectories of their dynamics often overlap in the Poincaré plot. However, AF are characterized by random behavior, thus, the trajectories of their dynamics in the Poincaré plot rarely overlap when compared to PAC/PVC. Therefore, the standard deviation of vector angles for PAC/PVC will be smaller than that for AF. We have heuristically determined that the standard deviation of the vector angles needs to be less than 10 degrees to be considered true PAC/PVC. Our choice of 10 degree threshold value will be further described and validated in the Results Section 3.3 and Figure 5a.

If the standard deviation of the vector angles is more than 10 degrees, the segment is no longer considered a candidate for PAC/PVC, and undergoes another test. The second threshold we examined is the ratio of the number of heart beat points in the zero quadrant to those that are outside it. The zero quadrant represents the change of heart rates for both the current and next beats that are less than 10 BPM. Since AF has highly varying heart rates, it will have more heart beat trajectory points that fall outside the zero quadrant than will NSR. For NSR, most of the heart beat points do reside inside the zero quadrant, provided that heart rates are unaffected by motion artifacts. If the ratio is less than 13% (heuristically determined), the initial determination that the segment includes AF is kept; otherwise, AF detection is replaced with NSR. As shown in Figure 3, NSR rhythms are typically contained within the zeroth quadrant whereas AF trajectories are rather highly spread and span out to other quadrants. Our choice of the 13% threshold value will be further described and validated in the Results Section 3.3 and Figure 5b.
**Algorithm 1** Algorithm for PAC/PVC Detection on AF Windows**Input:** $\theta_{123}:=$ degrees of included angle for ‘1-2-3’ pattern  $\theta_{645}:=$ degrees of included angle for ‘6-4-5’ pattern  $n_{123}:=$ number of ‘1-2-3’ pattern  $n_{645}:=$ number of ‘6-4-5’ pattern  $r_{zero}:=$ ratio of zero quadrants.**Output:** Arrhythmia type as NSR, PAC/PVC, or AF.  1: **if**
$(n_{123}+n_{645})\geq 3$
**then**  2: Calculate $std_{123}$ and $std_{645}$, the standard deviation of $\theta_{123}$ and $\theta_{645}$, respectively.  3: **if**
$n_{645}\leq 1$
**then**  4:  Set $std_{645}=0$.  5: **end if**  6: **if**
$n_{123}\leq 1$
**then**  7:  Set $std_{123}=0$.  8: **end if**  9: **if**
$std_{123}\leq 10$ && $std_{645}\leq 10$
**then**10:  Arrhythmia=PAC/PVC.11: **else**12:  **if**
$r_{zero}<0.13$
**then**13:   Arrhythmia=AF.14:  **else**15:   Arrhythmia=NSR.16:  **end if**17: **end if**18: **else**19: **if**
$r_{zero}<0.5$
**then**20:  Arrhythmia=AF.21: **else**22:  Arrhythmia=NSR.23: **end if**24: **end if**25: **return**
Arrhythmia

#### 2.2.4. PAC/PVC Detection for Non-AF Subjects

Algorithm 2 summarizes our proposed PAC/PVC detection algorithm for NSR detected segments. Similar to PAC/PVC detection for AF subjects, we examine if there are at least three or more ‘1-2-3’ or ‘6-4-5’ quadrant patterns. If there are three or more kite shaped patterns and the standard deviation of the vector angles is less than 10 degrees, the 30-s data segment is noted to contain PAC/PVC rhythms. If the standard deviation of the vector angles is greater than 10 degrees, the segment is further subjected to the following procedures.
**Algorithm 2** Algorithm for PAC/PVC Detection on Non-AF Windows**Input:** $\theta_{123},\theta_{645}:=$ degrees of included angle for ‘1-2-3’ and ‘6-4-5’ pattern, respectively. $\theta_{24},\theta_{42},\theta_{12},\theta_{23},\theta_{64},\theta_{45}:=$ degrees of slope angle for ‘2-4’,‘4-2’,‘1-2’,‘2-3’,‘6-4’,‘4-5’ pattern, respectively $n_{123},n_{645},n_{24},n_{42},n_{12},n_{23},n_{64},n_{45}:=$ number of ‘1-2-3’,‘6-4-5’,‘2-4’,‘4-2’,‘1-2’,‘2-3’,‘6-4’‘4-5’ pattern, respectively. $r_{zero}:=$ ratio of zero quadrants $n_{outside}:=$ number of points outside cross demarcation**Output:** Arrhythmia type as NSR, AF, or PAC/PVC.  1: **if**
$(n_{123}+n_{645})\geq 3$
**then**  2: Calculate $std_{123}$ and $std_{645}$, the standard deviation of $\theta_{123}$ and $\theta_{645}$, respectively.  3: **if**
$n_{645}\leq 1$
**then**  4:  Set $std_{645}=0$.  5: **end if**  6: **if**
$n_{123}\leq 1$
**then**  7:  Set $std_{123}=0$.  8: **end if**  9: **if**
$std_{123}\leq 10$ && $std_{645}\leq 10$
**then**10:  Arrhythmia=PAC/PVC.11: **else**12:  Arrhythmia=AF.13: **end if**14: **else**15: Calculate $std_{24}$ and $std_{42}$, the standard deviation of $\theta_{24}$ and $\theta_{42}$, respectively.16: **if**
$min(n_{24},n_{42})\geq 5$ && $std_{24}\leq 10$ && $std_{42}\leq 10$
**then**17:  Arrhythmia=PAC/PVC.18: **else**19;  Calculate $std_{12},std_{23},std_{64},std_{45}$, the standard deviation of $\theta_{12}$, $\theta_{23}$, $\theta_{64}$, and $\theta_{45}$, respectively.20:  **if**
$min(n_{64},n_{45})\leq 3$
**then**21:   Set $std_{64}=0$ and $std_{45}=0$.22:  **end if**23:  **if**
$min(n_{12},n_{23})\leq 3$
**then**24:   Set $std_{12}=0$ and $std_{23}=0$.25:  **end if**26:  **if**
$max(std_{12},std_{23},std_{64},std_{45})\leq 10$
**then**27:   Arrhythmia=PAC/PVC.28:  **else**29:   **if**
$r_{zero}<0.5\left|\right|n_{outside}\geq 5$
**then**30:    Arrhythmia=AF.31:   **else**32:    Arrhythmia=NSR.33:   **end if**34:  **end if**35: **end if**36: **end if**37: **return**
Arrhythmia

For NSR-detected data segments, we also look for the ‘2-4’ patterns since they represent bigeminy. To ensure that there are several PAC/PVC rhythms in a given 30-s data segment and to minimize false positive detection of PAC/PVC due to either poor signal-to-noise ratio or random artifacts, we have criteria that there needs to be more than 5 pairs of ‘2-4’ and ‘4-2’ patterns and the standard deviation of slope angles must be equal to or less than 10 degrees, to be considered a PAC/PVC segment. If neither of the above criteria is satisfied, we also check for other ‘kite’ shaped patterns.

For the two possible ‘kite’ shaped patterns associated with PAC/PVC, we subdivide ‘1-2-3’ and ‘6-4-5’ patterns into four vectors: vector ‘1-2’ and ‘2-3’ for pattern ‘1-2-3’, and ‘6-4’ and ‘4-5’ for pattern ‘6-4-5.’ The rationale for this is that ‘1-2-3’ or ‘6-4-5’ patterns might be interrupted by either bigeminy or trigeminy beats, which can result in the following patterns: ‘6-4-2-4-5’ or ‘6-4-0-5’. Thus, if we detect more than 3 pairs of either ‘1-2’ and ‘2-3’ or ‘6-4’ and ‘4-5,’ and the standard deviations of the paired vectors (e.g., combinations of either 1-2 or 2-3, and the same for 6-4 or 4-5) is less than 10 degrees, the data segment is then considered to contain PAC/PVC. Our choice of the 10 degree threshold value will be further described and validated in the Results Section 3.3 and Figure 5a.

If the standard deviation of the vector angle is greater than 10 degrees, we determine two criteria. First, we calculate the same threshold value as in Algorithm 1, the ratio of points that fall inside the zero quadrant to the entire number of datapoints in the Poincaré plot. The threshold for the ratio of points inside the zero quadrant was set to greater than 50%. Our choice of this threshold value will be further described and validated in the Results Section 3.3 and Figure 5c.

Second, we calculate the total number of points that fall outside the ‘cross’ demarcation (the area that quadrants zero, one, three, five, and six cover in the Poincaré plot). This threshold was set to further discriminate AF segments from those of PAC/PVC, so this case pertains to the case when AF was detected using our AF detection algorithm. The rationale for this is that for AF, the number of Poincaré trajectories that fall outside of the cross demarcation boundaries will be much larger than for PAC/PVC. As shown in Poincaré plots of Figure 3-(2) and (3) for AF and PAC/PVC, we observe many more trajectories that fall outside of the cross demarcation for the former. Hence, we set this threshold value to less than 5 points that all outside for the NSR detection decision for a given segment; otherwise, the NSR decision is reclassified as AF. Our choice of this threshold value will be further described and validated in the Results Section 3.3 and Figure 5d.

## 3. Results

Human adjudication of motion noise deemed that 292 30-s segments from 35 subjects were clean data. Our motion and noise artifact detection algorithm detected clean data segments from 37 subjects; 314 30-s segments were detected as clean [40]. The total number of 30-s segments from the 37 subjects is 2728, thus the coverage of detected clean window is 10.70%. Most of the clean data are from non-movement stage in our protocol like sitting or standing stage, and non-movement stage consists of only 35% of our experiment protocol in the ideal case. Moreover, for some subjects, their health conditions do not allow them to follow the protocol rules strictly, thus we have fewer clean windows than expected. The motivation to design an experiment protocol which includes different types of daily activities was to test if AF and PAC/PVC detection can be accurately performed on the smartwatch PPG data under motion noise artifacts. When compared to human adjudication of motion noise, our motion and noise artifact detection algorithm detected an additional 25 segments as clean and misdetected three segments as clean. The adjudication of the clean data and the determination that the algorithm had misdetected clean data was performed by two of the authors (KHC and DH). If there were disagreements between the two PPG readers on the results of adjudication for motion noise, a third person’s (SB) opinion was solicited and the majority view was selected as the ground truth. The same procedures were used for adjudication of the types of arrhythmias based on the reference ECG data, except that the third person (DM) is an electrophysiologist from the UMMC team.

Evaluation of our PAC/PVC detection algorithm is based on data from both the motion artifact detection algorithm (314 segments) and human adjudication of motion noise (292 segments). We show both data as this allows the results to be compared to our automated approach versus the human adjudication approach. Moreover, this allows us to examine how including 25 additional motion and noise corrupted data segments and the misdetected three clean segments affects the results when compared to human adjudication of motion noise.

### 3.1. Results of PAC/PVC Detection

First, we show results from the 314 segments that were detected as clean from our motion artifact detection algorithm [40]. The ground truth numbers based on human adjudication of motion noise, including the 25 additional misdetected segments by the motion artifact algorithm, are provided in the first row of Table 5. We adjudicated that there are 213, 55, and 46 segments associated with NSR, AF, and PAC/PVC, respectively. The confusion matrix depicting results of the types of rhythms without and with our proposed PAC/PVC detection algorithm is shown in Table 5. The purpose of displaying Table 5 is to quantify the number of PAC/PVC segments that are incorrectly labeled as either NSR or AF when our PAC/PVC detection algorithm is not used. Without the PAC/PVC detection algorithm, NSR and AF were detected in 248 and 52 segments, respectively. Note that PAC/PVC segments are classified as NSR in this case. The PAC/PVC detection algorithm, on the other hand, correctly classified 31 of the 46 as PAC/PVC and reduced 3 segments initially incorrectly detected as NSR to only 1.

Table 5 shows the performance evaluation of our PAC/PVC detection algorithm in more detail. The sensitivity, specificity, positive predictive value (PPV), negative predictive value (NPV) and accuracy for PAC/PVC label detection are 67.39%, 97.34%, 81.58%, 94.46% and 92.88%, respectively.

Table 5 also shows the results without our proposed PAC/PVC algorithm in the left two columns. The sensitivity of NSR classification values dropped slightly after our PAC/PVC detection, from 95.75% to 94.84%. However, the specificity, PPV, NPV and accuracy increased from 94.55% to 99.0%, 98.80% to 99.51%, 82.54% to 90.00%, and 95.54% to 96.17%, respectively. In addition, all five metrics’ values improved for AF classification after the PAC/PVC algorithm was used, as the sensitivity, specificity, PPV, NPV and accuracy all increased—from 94.55% to 98.18%, 95.75% to 97.90%, 82.54% to 91.53%, 98.80% to 99.57%, and 95.54% to 97.95%, respectively.

We also evaluated our arrhythmia classification algorithm results based on only the human adjudication of motion noise. Table 6 shows the results using the 292 segments manually-adjudicated to be clean. In this case, we found there to be 196, 54, and 42 segments associated with NSR, AF, and PAC/PVC, respectively. Without the PAC/PVC detection algorithm, NSR and AF were detected in 231 and 51 segments, respectively. With the PAC/PVC detection algorithm, it correctly classified 29 of the 42 as PAC/PVC and reduced the three segments initially incorrectly detected as NSR to only 1.

Table 6 shows the performance evaluation of our PAC/PVC detection algorithm in more detail. The sensitivity, specificity, positive predictive value, negative predictive value and accuracy for PAC/PVC label detection are 69.05%, 97.98%, 85.29%, 94.92% and 93.79%, respectively.

Table 6 also shows the results without our proposed PAC/PVC algorithm in the left two columns. The sensitivity of NSR classification values dropped slightly after our PAC/PVC detection, from 97.06% to 96.94%. However, the specificity, PPV, NPV, and accuracy increased from 94.44% to 98.95%, 87.93% to 94.00%, 98.72% to 99.48%, and 96.58% to 97.59%, respectively. In addition, all five metric values improved for AF classification after PAC/PVC was algorithm was used, where the sensitivity, specificity, PPV, NPV and accuracy all increased from 94.44% to 98.15%, 97.06% to 99.10%, 87.93% to 96.36%, 98.72% to 99.55%, and 96.58% to 98.91%, respectively.

### 3.2. Results of PAC/PVC Detection on Testing Datasets

We show the results of PAC/PVC detection using two additional testing datasets–MIMIC III and Gear S3 smartwatch data. We want to stress that no further tuning of the algorithm was done with these two datasets to detect PAC/PVC. The training of the algorithm was solely based on Simband smartwatch data.

#### 3.2.1. Gear S3 Dataset

The selected 101 30-s segments shown in Table 7 were adjudicated to determine that both the reference ECG and PPG signals were devoid of motion artifact. Among 101 30-s segments, we adjudicated that there are 54, 20, and 27 segments associated with NSR, AF, and PAC/PVC, respectively. The sensitivity, specificity, NPV, PPV, and accuracy of PAC/PVC classification were found to be 92.59%, 98.65%, 96.15%, 97.33%, and 97.03%, respectively. For AF classification, our results were 100% accurate, and for NSR classification, since we misdetected one NSR segment as PAC/PVC, the sensitivity, NPV, and accuracy for NSR classification are 98.15%, 97.92%, and 99.01%. The specificity and PPV for NSR classification were both 100%.

#### 3.2.2. MIMIC III Dataset

The results from 2614 30-s segments are shown in Table 8. The data consisted of 2220 segments of NSR, 311 of AF, and 83 of PAC/PVC. The sensitivity, specificity, NPV, PPV, and accuracy of PAC/PVC classification were found to be 92.77%, 100%, 100%, 99.76%, and 99.77%, respectively. For AF and NSR classification results, the sensitivity of AF is 94.86%, and the specificity of NSR is 95.94%. However, since there was only one false positive detection of AF, the specificity, PPV, NPV, and accuracy of AF were found to be 99.96%, 99.66%, 99.31%, and 99.35%, respectively. For NSR classification, since there was no false negative detection, the sensitivity, PPV, NPV, and accuracy were found to be 100%, 99.28%, 100%, and 99.39%, respectively.

### 3.3. Performance of PAC/PVC Detection with Difference Thresholds

For Algorithms 1 and 2, there are mainly three sets of threshold values. As shown in Figure 5a, the selected standard deviation of the vector angle threshold value of 10 degrees provides the best accuracy for the classification of the three classes (AF, NSR, and PAC/PVC), compared to other threshold values. Furthermore, shown in Figure 5a, increasing the vector angle higher than 10 degrees did not result in significantly better results. For example, 15 degrees provided an additional 0.01% accuracy for the detection of AF and an additional 0.32% accuracy for PAC/PVC class. We do note, however, that threshold values less than 10 degrees did result in significantly less accurate detection of PAC/PVC, although both AF and NSR classification results were less affected. Hence, while these results may vary slightly with additional new data, we believe that our choice of the vector angles’ standard deviation threshold value of 10 degrees is valid and that it may also be valid for new data.

The second threshold we examined is the ratio of heart beat points that fall in the zero quadrant of the Poincaré plot. For this threshold, which looks for the ratio of the number of points that reside within the zero quadrant to the total number of points in the Poincaré plot space, we have set two different thresholds to differentiate PAC/PVC from the algorithm-detected AF, and the algorithm-detected non-AF labels. The former is for Algorithm 1 and the latter is for Algorithm 2. Note that Algorithm 1 is for those heart beats that were initially detected as AF based on our AF detection algorithm. To ensure that frequent occurrences of PAC/PVC did not lead to false positive detection of AF, we examined the threshold values to determine whether or not the heartbeats in the 30-s segment were PAC/PVC. Algorithm 2 is valid only for the case when the AF detection algorithm states that the data segment contains non-AF rhythms. However, we evaluated Algorithm 2 to examine whether or not data segments contain PAC/PVC and discriminate these from NSR. Algorithm 1’s threshold value of 13% refers to the case when there are more than three occurrences of kite-shaped trajectories, but if not, a 50% threshold value was used. To examine the choice of these two threshold values, we varied these numbers, as shown in Figure 5b,c. The results are largely invariant to the threshold value ranges shown (0.05–0.2 for the first threshold value and 0.3–0.7 for the second threshold value). However, our chosen 13% and 50% values are the most optimal threshold values, as shown in these figures. Note that the 50% threshold value for Algorithm 2 was also chosen to discriminate PAC/PVC from NSR.

The third threshold is the number of heart beats that fall outside the cross-shaped demarcation boundaries in the Poincaré plot. To examine our choice of this threshold value of 5, we varied it from 1 to 15, as shown in Figure 5d. Similar to the prior threshold case (zero demarcation boundary), the results are largely invariant to the choice of threshold value. However, the most optimal value is 5.

Overall, the performance of our proposed algorithm did not change much when these three sets of thresholds values were varied. This is good news, as it instills confidence that our choice of these threshold values will most likely be valid for new data.

## 4. Discussion

The clinical importance of PAC/PVC detection is that frequent occurrences of these arrhythmia can mimic the behavior of AF when irregular pulse intervals are the main feature used to detect AF. Hence, the practical and ultimate aim of our PAC/PVC detection is to obtain more accurate AF detection using smartwatch PPG signals. Indeed, we have shown that detecting PAC/PVC can lead to significant false positive reduction during AF detection [40]. Other recent published reports [14,16,17,18,19,20,21,22,23,24,25,26,27,28,29,30,31,32] have largely focused on AF without accounting for PAC/PVC and consequently their accuracy of AF detection was suboptimal.

In this work, we illustrated a series of algorithmic approaches to discriminate PAC and PVC from the NSR and AF data segments. We have shown that our PAC/PVC detection algorithm improved the sensitivity and specificity values by 3.63% and 2.09%, respectively, for NSR detected segments. In addition, we correctly detected six out of seven PAC/PVC segments for those data initially misdetected as AF. Thus, the sensitivity for the PAC/PVC detection for AF was found to be 85.71%.

### 4.1. PAC/PVC Results Improved After Removing Noisy Windows

Comparing PAC/PVC detection results in Table 5 and Table 6, we can see that after manually removed those false positive detect of noisy segments, the sensitivity, specificity, PPV, NPV and accuracy are all increased. Though the values did not increase too much since our noise detection [40] is already accurate enough, our PAC/PVC detection could be improved if PPG noise detection is 100% accurate.

### 4.2. Limitations of the Poincaré Plot Approach to Detect PAC/PVC

We were limited by the number of subjects with PAC/PVC arrhythmias. However, the Poincaré plot approach proposed in this work is able to consistently reveal the distinct kite shape associated with PAC/PVC, provided that the PPG signal is relatively devoid of motion artifacts. Hence, the first criterion of the proposed approach to detect PAC and PVC starts with clean data segments. The accuracy of the proposed approach hinges on clean PPG data. How motion artifacts affect the accuracy of PAC/PVC detection is shown in Figure 6. Note that in Figure 6, though there exists high frequency noises in the PPG signal, our motion noise artifact detection algorithm still detects these segments as clean, which is good on increasing coverage but may induce false PAC/PVC detection [40]. All the PPG signals in Figure 6 exhibit high frequency noise. Our peak detection algorithm is affected by these high frequency noise artifacts as there are several instances where PPG heart rates deviate from the reference ECG heart rates. In the top right panel of Figure 6, a kite shape is formed in the Poincaré plot for NSR heart rates, which resembles the PAC/PVC patterns albeit we do not see the repeated kite shaped patterns associated with clean data. Certainly, having a criterion of repeated kite pattern can lead to false negative PAC/PVC detection in these noise-corrupted PPG data. The same issue arises for row 3 of Figure 6, which represents a noise-contaminated PAC signal. Noise-artifact effect is seen in the middle portion of the 30-s data segment, which results in an atypical quadrant pattern ‘4-2-3.’ Based on this atypical pattern, our algorithm will not detect this data segment to contain PAC/PVC rhythms and consequently leads to false negative detection.

Another instance of low sensitivity for PAC/PVC detection arises from atypical patterns of PAC/PVC, albeit these instances are less frequent. An example of atypical beats occurs at around time 2890 s to 2895 s in the middle set of panels in Figure 6. For this data segment, there are three consecutive fast PAC beats, but the Poincaré plot trajectory reveals a ‘5-0-6’ pattern with no distinct kite shaped pattern, leading to false negative detection. Cases such as these can potentially benefit from combining image-processing techniques with machine learning methods to overcome obstacles associated with accurate detection of PAC/PVC, especially when data are contaminated with noise artifacts.

### 4.3. Signal Window Length Limited PAC/PVC Detection Accuracy

In this study, the data segment size was limited to 30 s for the purpose of possible real-time detection. However, this is a challenging scenario as PAC/PVC occurrences within a 30-s data segment tend to be limited to a few beats. Increasing the data length requirement to at least 1 minute will most likely increase the sensitivity of PAC/PVC detection. The standardization of the number of PAC/PVC patterns within a certain data length segment to be considered accurate detection has not yet been established by the cardiac rhythm community.

### 4.4. Limitation of Small Sample Size

As this is a preliminary study to design an algorithm for detection of PAC/PVC, the data samples were not large. To compensate for a limited data sample size, we have tested the algorithm using two independent datasets with two different sensor modalities. The results were good, as accuracies were higher than 92% for the two independent test datasets. Note that these results were based on fixing all parameters derived from the training datasets from Simband and no further tuning was done for the testing datasets from MIMIC III and Gear S3. Thus, these new results provide further verification of the proposed PAC/PVC detection method. It is clear that if unknown patterns associated with PAC/PVC occur in new data, certainly our method’s accuracy will suffer accordingly. Thus, it remains to be seen if our results will remain as accurate with more data.

## 5. Conclusions

In this study, we introduced a novel approach using the Poincaré plot and vector resemblance method to perform PAC/PVC detection on smartwatch PPG data from AF and NSR subjects. Our proposed approach has been shown to accurately distinguish PAC/PVC occurrences from subjects with either NSR or AF. The ultimate benefit of this capability is not only the detection of PAC/PVC, but fewer false positive detections of AF from NSR rhythms. This is because frequent occurrences of PAC/PVC beats in a given NSR rhythm can lead to incorrect detection of AF, as these arrhythmias introduce some randomness in the variability of heart rates. The independent test datasets using different sensor modalities (Gear S3 smartwatches and clinical pulse oximeter-derived PPG MIMIC III database) showed that our proposed method provides rather high detection accuracy (>92% on both datasets). These results instill some confidence in the generalizability of the proposed method.

## Figures and Tables

**Figure 1 sensors-20-05683-f001:**
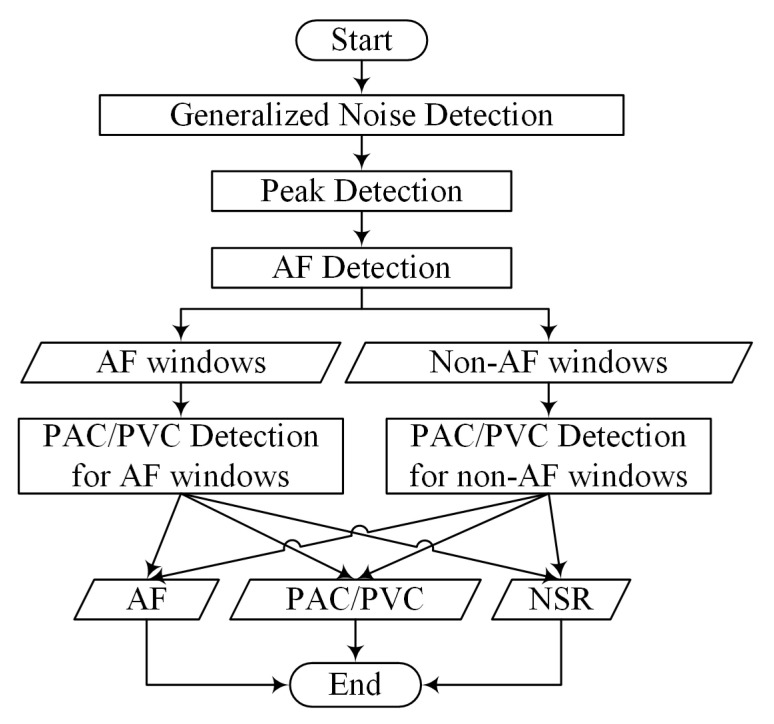
Flow chart of entire AF detection and PAC/PVC detection system.

**Figure 2 sensors-20-05683-f002:**
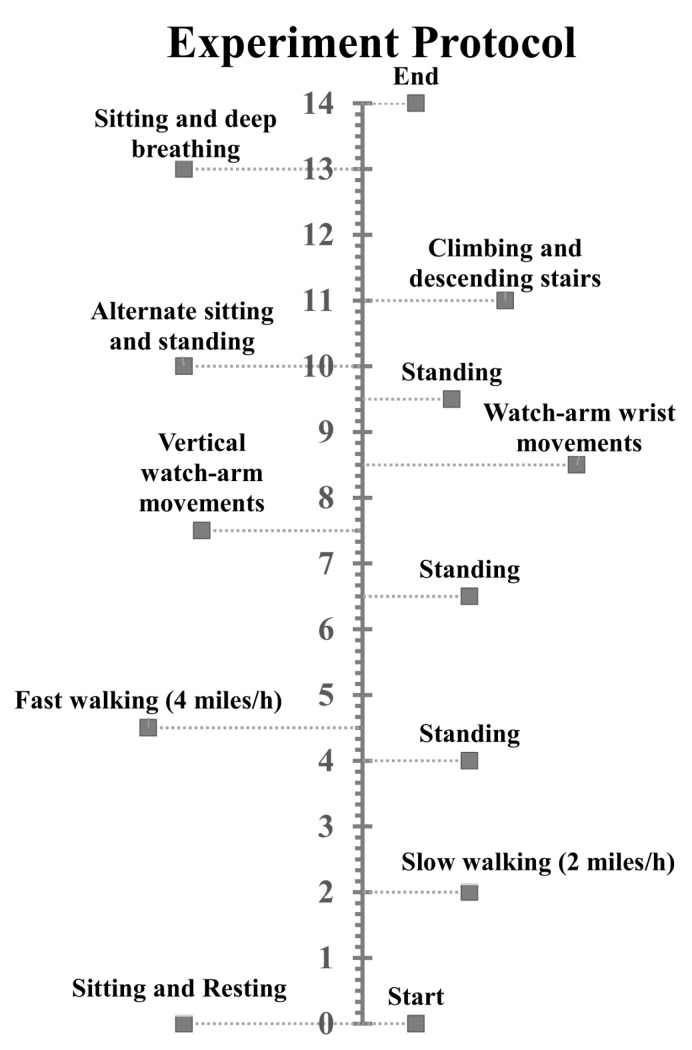
Timeline of the experimental protocol (unit: minute).

**Figure 3 sensors-20-05683-f003:**
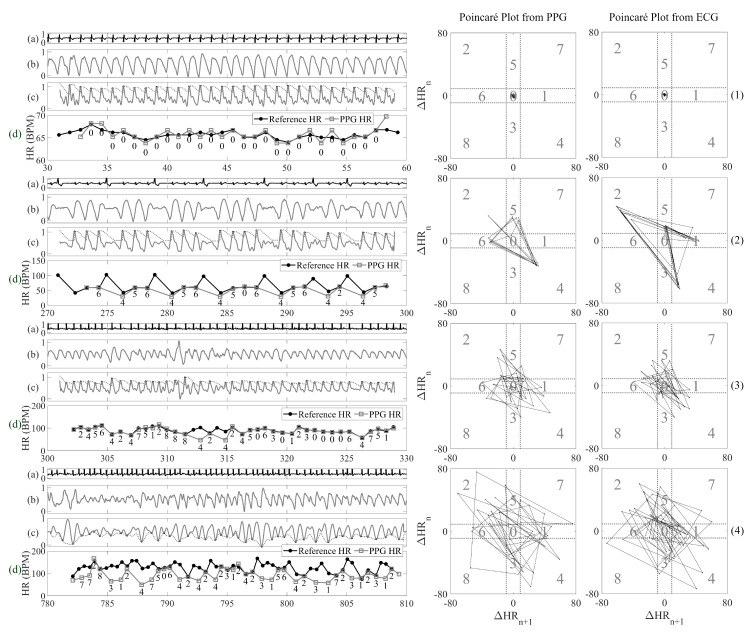
Poincaré plots of PPG HR and reference ECG HR (middle panels and right panels) for NSR, PVC, normal heart rate AF and fast heart rate AF for the data shown in the left panels (1), (2), (3) and (4), respectively. Each panel of the left column shows (**a**): reference ECG, (**b**) PPG, (**c**) detected peaks from the filtered PPG and the (**d**) reference ECG HR and PPG HR from a 30-s segment.

**Figure 4 sensors-20-05683-f004:**
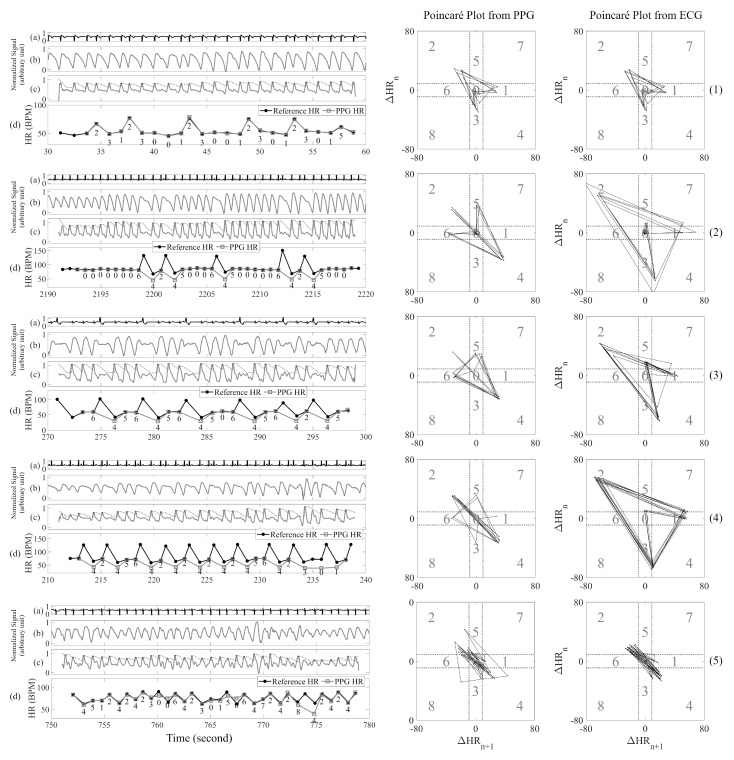
PAC/PVC Poincaré plot patterns. Panel (1) is ‘1-2-3’ pattern for normal heart rate PAC; panel (2) is ‘6-4-5’ pattern for fast heart rate PAC; panel (3) is ’6-4-5’ pattern for fast heart rate PVC; panel (4) is ‘2-4’ pattern for fast heart rate PAC with trigeminy; panel (5) is ‘2-4’ pattern for normal heart rate PAC with bigeminy.

**Figure 5 sensors-20-05683-f005:**
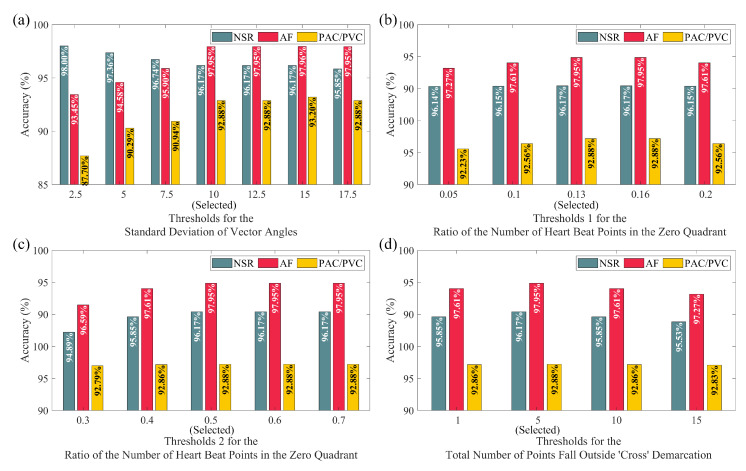
Performance of the proposed algorithms with different thresholds based on automatic noise detection using Simband Dataset: accuracy of NSR, AF, and PAC/PVC classification with (**a**) thresholds for the standard deviation of vector angles; (**b**,**c**) two thresholds for the ratio of the number of heart beat points in the zero quadrant; (**d**) thresholds for the total number of points fall outside the ’cross’ demarcation.

**Figure 6 sensors-20-05683-f006:**
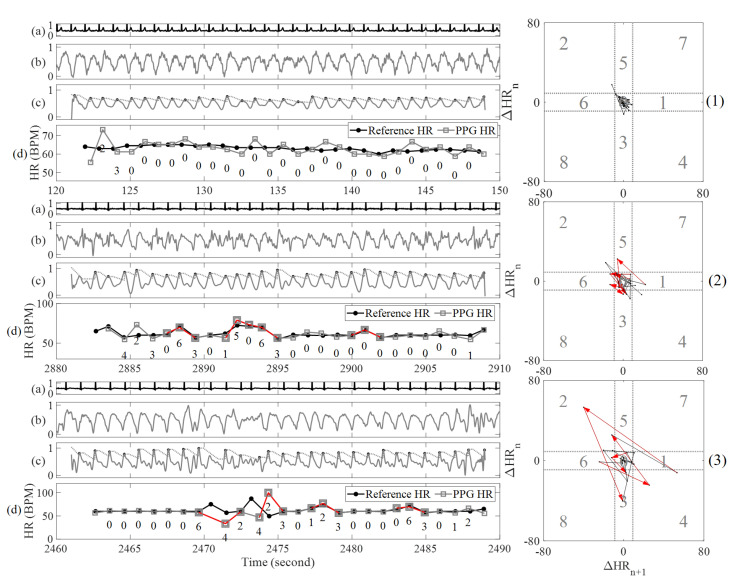
Difficulties in PAC/PVC detection using Poincaré Plot. Panel (1) is from noisy NSR subject (4044); panel (2) is from noisy PAC subject (4017) with atypical PAC pattern; panel (3) is from same noisy PAC subject (4017).

**Table 1 sensors-20-05683-t001:** Detailed subject information of training dataset—UMMC Simband dataset.

				Adjudicated					Detected			
Subject ^1^	Age	Sex	Duration ^2^	Clean (%) ^3^	NSR (%) ^4^	AF (%)	PAC (%)	PVC (%)	NSR (%) ^5^	AF (%)	PAC (%)	PVC (%)
4001	66	male	47	13 (27.66%)	0	0	13 (100.00%)	0	0	0	10 (76.92%)	0
4002	58	male	44	12 (27.27%)	12 (100.00%)	0	0	0	12 (100.00%)	0	0	0
4004	76	female	47	0	0	0	0	0	0	0	0	0
4005	54	male	60	4 (6.67%)	4 (100.00%)	0	0	0	4 (100.00%)	0	0	0
4006	69	male	46	3 (6.52%)	0	3 (100.00%)	0	0	0	3 (100.00%)	0	0
4007	91	male	103	10 (9.71%)	0	10 (100.00%)	0	0	0	10 (100.00%)	0	0
4008	62	female	44	2 (4.55%)	2 (100.00%)	0	0	0	2 (100.00%)	0	0	0
4009	59	male	45	0	0	0	0	0	0	0	0	0
4012	72	male	65	7 (10.77%)	0	7 (100.00%)	0	0	0	6 (85.71%)	0	0
4013	50	male	27	12 (44.44%)	0	12 (100.00%)	0	0	0	12 (100.00%)	0	0
4015	74	male	43	2 (4.65%)	0	2 (100.00%)	0	0	0	2 (100.00%)	0	0
4016	62	male	36	3 (8.33%)	0	3 (100.00%)	0	0	0	3 (100.00%)	0	0
4017	76	male	98	15 (15.31%)	8 (53.33%)	0	7 (46.67%)	0	7 (87.50%)	0	0	0
4019	60	male	53	3 (5.66%)	3 (100.00%)	0	0	0	3 (100.00%)	0	0	0
4020	69	male	48	3 (6.25%)	3 (100.00%)	0	0	0	3 (100.00%)	0	0	0
4021	77	female	49	15 (30.61%)	6 (40.00%)	0	9 (60.00%)	0	6 (100.00%)	0	7 (46.67%)	0
4022	63	male	87	10 (11.49%)	1 (10.00%)	0	9 (90.00%)	0	1 (100.00%)	0	8 (80.00%)	0
4024	70	male	133	2 (1.50%)	0	2 (100.00%)	0	0	0	2 (100.00%)	0	0
4025	64	male	110	3 (2.73%)	3 (100.00%)	0	0	0	3 (100.00%)	0	0	0
4026	74	male	106	4 (3.77%)	4 (100.00%)	0	0	0	4 (100.00%)	0	0	0
4027	73	male	100	31 (31.00%)	31 (100.00%)	0	0	0	30 (96.77%)	0	0	0
4028	69	male	63	6 (9.52%)	6 (100.00%)	0	0	0	6 (100.00%)	0	0	0
4029	69	male	143	18 (12.59%)	18 (100.00%)	0	0	0	18 (100.00%)	0	0	0
4030	74	female	68	18 (26.47%)	18 (100.00%)	0	0	0	18 (100.00%)	0	0	0
4032	67	female	92	2 (2.17%)	2 (100.00%)	0	0	0	2 (100.00%)	0	0	0
4033	82	male	95	39 (41.05%)	39 (100.00%)	0	0	0	39 (100.00%)	0	0	0
4034	70	female	88	8 (9.09%)	7 (87.50%)	0	1 (12.50%)	0	6 (85.71%)	0	1 (12.50%)	0
4035	88	male	29	2 (6.90%)	2 (100.00%)	0	0	0	2 (100.00%)	0	0	0
4036	75	female	88	1 (1.14%)	1 (100.00%)	0	0	0	1 (100.00%)	0	0	0
4037	69	male	46	3 (6.52%)	0	0	0	3 (100.00%)	0	0	0	3 (100.00%)
4038	81	male	98	11 (11.22%)	0	11 (100.00%)	0	0	0	11 (100.00%)	0	0
4040	74	male	120	2 (1.67%)	2 (100.00%)	0	0	0	2 (100.00%)	0	0	0
4041	81	male	89	3 (3.37%)	3 (100.00%)	0	0	0	1 (33.33%)	0	0	0
4042	76	male	74	9 (12.16%)	9 (100.00%)	0	0	0	8 (88.89%)	0	0	0
4043	75	male	71	4 (5.63%)	0	4 (100.00%)	0	0	0	4 (100.00%)	0	0
4044	70	male	94	3 (3.19%)	3 (100.00%)	0	0	0	3 (100.00%)	0	0	0
4045	69	female	79	9 (11.39%)	9 (100.00%)	0	0	0	9 (100.00%)	0	0	0
Total	70.49	21.62% female	2728	292 (10.70%)	196 (67.12%)	54 (18.49%)	39 (13.36%)	3 (1.03%)	190 (96.94%)	53 (98.15%)	26 (66.67%)	3 (100.00%)
^1^ Discontinuity in the subjects’ IDs occurs there was no interpretable data recorded from those subjects; ^2^ Monitoring Duration (segments); ^3^ Percentage (%) is based on monitoring duration; ^4^ Arrhythmia burden (%) is based on adjudicated clean segments; ^5^ Percentage (%) is based on Arrhythmia Burden.												

**Table 2 sensors-20-05683-t002:** Selected subject information in testing dataset: MIMIC III.

Subject Index	Subject ID in MIMIC III
AF1	p075796-2198-07-25-23-40
AF2	p026377-2111-11-17-16-46
NSR1	p004829-2103-08-30-21-52
NSR2	p013072-2194-01-22-16-13
NSR3	p050384-2195-01-30-02-21
NSR4	p055204-2132-06-30-09-34
NSR5	p058932-2120-10-13-23-15
PACPVC1	p007614-2177-01-08-13-21
PACPVC2	p065656-2121-08-07-04-35
PACPVC3	p030542-2135-10-17-10-33

**Table 3 sensors-20-05683-t003:** Poincaré plot quadrants associated with various heart beat arrhythmia permutation based on difference of three consecutive heartbeats.

${\mathrm{HR}}_{n-1}-{\mathrm{HR}}_{n}-{\mathrm{HR}}_{n+1}$	Quadrant Number
slow-slow-slow	0
fast-fast-fast	0
slow-slow-fast	1
slow-fast-slow	2
fast-slow-slow	3
fast-slow-fast	4
slow-fast-fast	5
fast-fast-slow	6
slow-fast-ultra fast	7
ultra fast-fast-slow	8

**Table 4 sensors-20-05683-t004:** Arrhythmias with their corresponding trajectory patterns in Poincaré plots.

Types of PAC/PVC	Trajectory Patterns
Bigeminy (normal HR)	2-4-2-4-2-4-…
Bigeminy (Fast HR)	0-0-0-0-0-0-…
Trigeminy (Fast HR)	2-4-2-4-2-4-…
Trigeminy (normal HR)	1-2-3-1-2-3-…
Quadrigeminy (fast HR)	6-4-5-6-4-5-…
Quadrigeminy (normal HR)	1-2-3-0-1-2-3-…

**Table 5 sensors-20-05683-t005:** Evaluation of proposed PAC/PVC detection method based on automatic noise detection.

	Without PAC/PVC Detection		With PAC/PVC Detection		
	Non-AF	AF	NSR	AF	PAC/PVC
Ground Truth	259	55	213	55	46
True Positive	248	52	202	54	31
True Negative	52	248	99	233	256
False Positive	3	11	1	5	7
False Negative	11	3	11	1	15
Sensitivity	95.75	94.55	94.84	98.18	67.39
Specificity	94.55	95.75	99.00	97.90	97.34
PPV	98.80	82.54	99.51	91.53	81.58
NPV	82.54	98.80	90.00	99.57	94.46
Accuracy	95.54	95.54	96.17	97.95	92.88

**Table 6 sensors-20-05683-t006:** Evaluation of proposed PAC/PVC detection method based on manual adjudication of clean windows.

	Without PAC/PVC Detection		With PAC/PVC Detection		
	Non-AF	AF	NSR	AF	PAC/PVC
Ground Truth	238	54	196	54	42
True Positive	231	51	190	53	29
True Negative	51	231	94	219	243
False Positive	3	7	1	2	5
False Negative	7	3	6	1	13
Sensitivity	97.06	94.44	96.94	98.15	69.05
Specificity	94.44	97.06	98.95	99.10	97.98
PPV	98.72	87.93	99.48	96.36	85.29
NPV	87.93	98.72	94.00	99.55	94.92
Accuracy	96.58	96.58	97.59	98.91	93.79

**Table 7 sensors-20-05683-t007:** Evaluation of proposed PAC/PVC detection method on independent gear S3 dataset.

	With PAC/PVC Detection		
	NSR	AF	PAC/PVC
Ground Truth	54	20	27
True Positive	53	20	25
True Negative	47	81	73
False Positive	0	0	1
False Negative	1	0	2
Sensitivity	98.15	100	92.59
Specificity	100	100	98.65
PPV	100	100	96.15
NPV	97.92	100	97.33
Accuracy	99.01	100	97.03

**Table 8 sensors-20-05683-t008:** Evaluation of proposed PAC/PVC detection method on independent MIMIC III dataset.

	With PAC/PVC Detection		
	NSR	AF	PAC/PVC
Ground Truth	2220	311	83
True Positive	2220	295	77
True Negative	378	2302	2531
False Positive	16	1	0
False Negative	0	16	6
Sensitivity	100	94.86	92.77
Specificity	95.94	99.96	100
PPV	99.28	99.66	100
NPV	100	99.31	99.76
Accuracy	99.39	99.35	99.77

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
