# Peer review of "Premature Atrial and Ventricular Contraction Detection Using Photoplethysmographic Data from a Smartwatch"

_sensors, 2020, doi:10.3390/s20195683_

Round 1

Reviewer 1 Report

In this work, the authors present a method to detect premature atrial and ventricular contraction (PAC/PVC) from smartwatch PPG data. The possibility to clearly identify these events should allow to improve the automatic detection of atrial fibrillation (AF) reducing the number of false positive.

The data presented, even if on a small number of subjects, seem confirming the capacity of the algorithm proposed to identify PAC/PVC and to improve the detection performance of AF (specificity, sensitivity and accuracy).

My main concern regards the thresholds eristically defined in the vector resemblance algorithm. It would be interesting to know:1) if there is any reason that drive you towards these thresholds, 2) if the performance of the algorithm changes in a significant way modifying these values.

Minor revisions are available as comments in the attached pdf.

Author Response

Reviewer 1:

We would like to thank the reviewer for the positive judgment of our work and for the remarks, which have led to the improvement of our paper. Below we include a reply to the reviewer’s comment. We have modified our paper following this comment and we hope that it is now suitable for publication. Please note that all changes to the text are highlighted in the revised manuscript.

Comments to the Corresponding Author:

In this work, the authors present a method to detect premature atrial and ventricular contraction (PAC/PVC) from smartwatch PPG data. The possibility to clearly identify these events should allow to improve the automatic detection of atrial fibrillation (AF) reducing the number of false positive.

The data presented, even if on a small number of subjects, seem confirming the capacity of the algorithm proposed to identify PAC/PVC and to improve the detection performance of AF (specificity, sensitivity and accuracy).

Major comments:

My main concern regards the thresholds eristically defined in the vector resemblance algorithm. It would be interesting to know:

1) if there is any reason that drive you towards these thresholds,

2) if the performance of the algorithm changes in a significant way modifying these values.

Thank you for your positive comments. For Algorithms 1 and 2, there are mainly three thresholds. The first threshold involves the standard deviation of vector angles. Standard deviation is a measure of the trajectory resemblance on Poincare plots. Because PAC/PVC occurrences are repeated (e.g., bigeminy, trigemini, etc.) the trajectories of their dynamics often overlap in the Poincare plot. However, AF are characterized by random behavior, thus, the trajectories of their dynamics in the Poincare plot rarely overlap when compared to PAC/PVC. Therefore, the standard deviation of vector angles for PAC/PVC will be smaller than that for AF. As shown in Figure 5-(a) in our revised paper, the selected standard deviation of the vector angle threshold value of 10 degrees provides the best accuracy for the classification of three classes (AF, NSR, and PAC/PVC), compared to other threshold values. Also shown in Figure 5-(a), increasing the vector angle higher than 10 degrees did not result in significantly better results. For example, 15 degrees provided additional 0.01% accuracy for the detection of AF and an additional 0.32% accuracy for PAC/PVC class. We do note, however, that threshold values less than 10 degrees did result in significantly less accurate detection of PAC/PVC although both AF and NSR classification results were less affected. Hence, while these results may vary slightly with additional new data, we believe that our choice of the vector angles’ standard deviation threshold value of 10 degrees is valid and that it may also be valid for new data.

The second threshold we examined is the ratio of heart beat points that fall in the zero quadrant of the Poincare plot. The zero quadrant represents the change of heart rates for both the current and next beats that are less than 10 BPM. Since AF has highly varying heart rates, it will have more heart beat trajectory points that will fall outside the zero quadrant than NSR. For NSR, most of the heart beat points do reside inside the zero quadrant, provided that heart rates are unaffected by motion artifacts. For this threshold, which looks for the ratio of number of points that reside within the zero quadrant to the total number of points in the Poincare plot space, we have set two different thresholds to differentiate PAC/PVC from the algorithm-detected AF, and the algorithm-detected non-AF labels. The former is for Algorithm 1 and the latter is for Algorithm 2 (pages 11-12 in the manuscript). Please note that Algorithm 1 is for those heart beats that were initially detected as AF based on our AF detection algorithm. To ensure that frequent occurrences of PAC/PVC did not lead to false positive detection of AF, we examined the threshold values to determine whether or not the hearts beats in the 30-sec segment is PAC/PVC. Algorithm 2 is valid only for the case when the AF detection algorithm states that the data segment contains non-AF rhythms. However, we evaluated Algorithm 2 to examine whether or not data segments contain PAC/PVC and discriminate these from NSR. Algorithm 1’s threshold values of 13% refers to the cases when there are more than 3 occurrences of kite-shaped trajectories, but if not, a 50% threshold value was used. To examine the choice of these two threshold values, we varied these numbers, as shown in Figure 5-(b) and (c). The results are largely invariant to the threshold values ranges shown (0.05-0.2 for the first threshold value and 0.3-0.7 for the second threshold value). However, our chosen 13% and 50% values are the most optimal threshold values, as shown in these figures. Note that the 50% threshold value for Algorithm 2 was also chosen to discriminate PAC/PVC from those of NSR.

The third threshold is the number of heart beats that fall outside the cross-shaped demarcation boundaries in Poincare plot. The ‘cross-shaped demarcation boundaries’ consists of the quadrants zero, one, three, five, and six. This threshold was set to further discriminate AF segments from those of PAC/PVC, so this case pertains to the case when AF was detected using our AF detection algorithm. The rationale for this is that for AF, the number of Poincare trajectories that fall outside of the cross-shape demarcation boundaries will be much larger than for PAC/PVC, hence, we set this threshold value to 5. As shown in Poincare plots of Figure 3-(2) and (3) for AF and PAC/PVC, we observe much more trajectories that fall outside of the cross demarcation for the former. To examine our choice of this threshold value of 5, we varied it from 1 to 15, as shown in Figure 5-(d). Similar to the prior threshold case (zero demarcation boundary), the results are largely invariant to the choice of threshold value. However, the most optimal value at 5.

Overall, the performance of our proposed algorithm did not change much when these three sets of thresholds values were varied. This is good news, as it instills confidence that our choice of these threshold values will most likely be valid for new data.

Minor revisions are available as comments in the attached pdf.

Page 1: NSR and AF acronym not defined;

Thank you. We’ve defined them when they first appeared on line 7.

Page 2: ‘As a significant percent of the population affected by AF are asymptomatic…,’ change ‘affected by’ to ‘suffering from’;

Thank you for your suggestion! We have corrected ‘affected by’ into ‘suffering from’ on line 33.

Page 3: Did you test this threshold in any way? Can you provide details about the rationale of this choice. Did you consider the SNR of the two systems?

Thank you for your comment. Please note that our choice of the correlation value is not to be considered as a threshold value. As the reviewer is aware, the cross-correlation is the most common way to align two ECG signals (please see reference [34]). When two signals are well-aligned, the cross-correlation value will have a value >0.8 and if they are not, it can be lower than 0.4. However, even for perfectly aligned signal, the cross-correlation value will be lower than 1 if they have a low signal-to-noise ratio. For this reason, we chose 0.9 for the alignment of ECG data from Simband and Holter monitors. We have modified the text from line 102 to line 108.

Page 4: Missing data in table 1.

Thank you for your comment. We have updated the age and sex information for 4009 in Table 1.

Page 4: Title of 2.1.2, ‘additional testing dataset…’ Because this is the first testing dataset you introduce, I would remove the adjective "Additional"

Thank you for your suggestions. We removed the ‘additional’ in the title of 2.1.2.

Page 4: how many patients did you enroll? Can you add a table detailing information about them?

We used two patients from our on-going clinical trial dataset. Subject 005 was non-AF with 54 NSR segments and 27 PAC/PVC segments. Subject 307 was an AF subject with 20 clean AF segments. Please note that we have used MIMIC III data from 2 AF and 3 PAC/PVC subjects which were used as testing data.

Page 4: Missing ‘s’ in ‘patients.’

We corrected ‘patient’ to ‘patients’ on line 124.

Page 4: Which were the selection criteria (for subjects you selected in MIMIC III dataset)?

Since the ICU recording for each subject in MIMIC III [38] dataset contained hundreds of hours of data, we only used the subjects whose data had already been prepared for the AF study in the reference [39], in which cardiologists had spent several months to adjudicate the presence of AF in those recordings. The AF detection study described in [39] was based on long-term ECG recordings, where the subjects were selected from critically ill sepsis patients. Since the PPG signal was neither analyzed nor observed in that study, only a few subjects in that study had simultaneous PPG and ECG signals. However, for this paper we needed subjects with simultaneous ECG and PPG recordings, as well as AF annotations from the cardiologists. As a result, we manually selected a random subset of 10 subjects from the study population of [39] for whom simultaneous ECG and PPG signals were available and used this subset of 10 subjects as a separate blind test dataset.

Page 10: Can you provide some details that drives you toward this assumption

Please see our reply to your major concern 1.

Page 11: Missing ‘s’ in ‘patterns.’

We have added the plural form of ‘patterns’ on line 296.

Reviewer 2 Report

The authors present a novel algorithm for detecting premature atrial/ventricular complexes as distinct from atrial fibrillation using photoplethymogram as measured using the Samsung Simband smartwatch. The algorithm for PAC/PVC detection improved the sensitivity, specificity, and accuracy of AF detection. I have the following comments:

  • The trajectory of both PAC and PVC appears the same, both '6-4-5' as a downward kite--does this mean the algorithm would not be able to distinguish between the two? I presume if the aim was to reduce false positive detection of AF, it would not be necessary to distinguish PAC and PVC?
  • It was unclear to me whether the Poincare plots from ECG were used as the reference. Also, it was interesting to see that the kite pattern were in opposite directions between the Poincare plots from PPG and the ones from ECG
  • In Table 5, numbers in the confusion matrix for non-AF didn't add up to 314
  • Table titles for Tables 3 and 4 were the same. Perhaps the authors meant for Table 4 to be titled trajectory associated with arrhythmias?
  • In Figures 3 and 4, panels 1c, 2c, 3c, there are two signals but they weren't indicated in the labels. I'm assuming one is the filtered PPG? Also, it appears that the troughs were being detected in 4c, and the peaks that were shown align with the troughs of the PPGs shown in b panels. 
  • The authors pointed out that the clean signals constituted only roughly 10% of all the recorded segments, and that most were from non-movement measurements. Perhaps this is worthy of further discussion in terms of applicability and future improvements for implementing the algorithm in future smartwatches. It was really helpful that the authors included the noisy signals to present the difficulties in PAC/PVC detection.
  • Minor comments
    • 2.1.1. UMass should be UMCC
    • Table 1 is slightly hard to read
    • the flow may read better if the experimental protocols are listed under the same subsections as description of the datasets
    • A brief description of the motion artefact detection could still be included to help readers understand what is being done
    • x-axis should be included in Figures 3-4 as per Figure 5

Author Response

Reviewer: 2

We would like to thank the reviewer for the positive judgment of our work and for the remarks, which have led to the improvement of our paper. Below we include a reply to the reviewer’s comment. We have modified our paper following this comment and we hope that it is now suitable for publication. Please note that all changes to the text are highlighted in the revised manuscript.

Comments to the Corresponding Author

The authors present a novel algorithm for detecting premature atrial/ventricular complexes as distinct from atrial fibrillation using photoplethymogram as measured using the Samsung Simband smartwatch. The algorithm for PAC/PVC detection improved the sensitivity, specificity, and accuracy of AF detection. I have the following comments:

Major comments:

           The trajectory of both PAC and PVC appears the same, both '6-4-5' as a downward kite--does this mean the algorithm would not be able to distinguish between the two? I presume if the aim was to reduce false positive detection of AF, it would not be necessary to distinguish PAC and PVC?

Thank you for your comment. The reviewer is correct. The aim of this paper was not to discriminate between PAC vs. PVC. The aim was to discriminate PAC/PVC from AF. We have an approach which appears to be successful in discriminating between PAC and PVC but due to a small number of subjects with PVC, we have refrained from including this result in our manuscript. Once we have more PVC subjects, we hope to further examine discrimination between PAC and PVC.

           It was unclear to me whether the Poincare plots from ECG were used as the reference. Also, it was interesting to see that the kite pattern were in opposite directions between the Poincare plots from PPG and the ones from ECG

We apologize for the confusion. Please note that we did not use the Poincare plots from ECG as the reference. We only used them to illustrate that similar to ECG, the kite-shaped trajectories in Poincare plots are also seen in PPG. We do note, however, that the upside down of the kite-shaped pattern in the Poincare plot of PPG are due to fast heart beats in PAC/PVC.

           In Table 5, numbers in the confusion matrix for non-AF didn't add up to 314

Thank you for noticing this error. We made this mistake in Table 5 because we did not add 7 PAC/PVC segments that were mistakenly detected as AF to ‘false negative (FN)’ calculation in ‘Non-AF’ class. We corrected this mistake as ‘FN’ of ‘non-AF’ class for both Tables 5 and 6. For Table 5, the sensitivity, NPV, and accuracy of non-AF class without PAC/PVC detection changed from 98.41% to 95.75%, 92.86% to 82.54%, and 97.72% to 95.54%, respectively, after accounting for this omission. In Table 6, the sensitivity, NPC, and accuracy changed from 99.57% to 97.06%, 98.08% to 87.93%, and 98.60% to 96.58%, respectively.

           Table titles for Tables 3 and 4 were the same. Perhaps the authors meant for Table 4 to be titled trajectory associated with arrhythmias?

Thank you for noting our error. We have corrected it to ‘Arrhythmias with their corresponding trajectory patterns in Poincare plots.’

           In Figures 3 and 4, panels 1c, 2c, 3c, there are two signals but they weren't indicated in the labels. I'm assuming one is the filtered PPG? Also, it appears that the troughs were being detected in 4c, and the peaks that were shown align with the troughs of the PPGs shown in b panels.

The reviewer is correct, so we have added the ‘filtered’ in the caption of Figure 3. All C panels in each figure show filtered PPG whereas unfiltered PPG are shown in the panel B. The reviewer is also correct that we have used troughs of the PPG for peak detection especially for AF and fast heart rates. For non-AF, we have used the peaks instead of troughs of the PPG.

           The authors pointed out that the clean signals constituted only roughly 10% of all the recorded segments, and that most were from non-movement measurements. Perhaps this is worthy of further discussion in terms of applicability and future improvements for implementing the algorithm in future smartwatches. It was really helpful that the authors included the noisy signals to present the difficulties in PAC/PVC detection.

Thanks for your suggestions and comments! We also agree with the reviewer that PPG data quality would be worse in real life daily monitoring. Since we need clean data to accurately detect PAC/PVC arrhythmia, we have been conservative in only assessing clean data segments that are noise-free (devoid of spikes or bad signal quality data) for the entire 30-sec. Otherwise, motion-corrupted data segments will lead to many false positive detections of either AF or PAC/PVC. However, in our on-going 14-day clinical trial, we found that some of our older subjects are less active than our experiment protocol which were largely from students. Moreover, we also found that more than 70% of the watch data were found to be clean and usable during the night time if the subjects properly wear the watch. Thus, we are optimistic with the proposed algorithm as well as the stringent motion artifact detection for smartwatch AF monitoring and PAC/PVC detection. We have added some more text to reflect the above.

Minor comments:

·                 2.1.1. UMass should be UMCC

Thank you for your suggestion! We have corrected it to ‘UMMC’ in title for 2.1.1 and the title of Table 1.

·                 Table 1 is slightly hard to read

Thank you for your suggestion! We cleaned up the header of each column in Table 1 to make it easier to read.

·                 the flow may read better if the experimental protocols are listed under the same subsections as description of the datasets

Thank you very much for your suggestion! We moved the content of 2.1.4 to 2.1.1 and the content of 2.1.5 under 2.1.2, and then deleted 2.1.4 and 2.1.5.

·                 A brief description of the motion artefact detection could still be included to help readers understand what is being done

Thank you for your suggestion! To briefly summarize the motion artifact detection [40]: we used two signals, the accelerometer (ACC) and PPG signals, to decide if motion noise existed in a 30-sec segment. Motion artifact detection for the ACC signal was based on a thresholding method on the mean value of the univariance ACC data. For detecting motion artifacts in the PPG signal, a time-frequency spectra (TFS) was first calculated from the univariance and zero-mean PPG, and then two signals’ power consisting of the clean (Q1) and noisy (Q2) signals, were calculated from the TFS. By counting the time instances of Q1 and Q2, we can separate the corrupted PPG signal from the clean PPG signal. We considered a 30-sec PPG segment as clean only when both the PPG and its corresponding ACC segment were detected as clean. We added above description in the end of the first paragraph of the section 2.2.1

·                 x-axis should be included in Figures 3-4 as per Figure 5

Thank you for your suggestion. However, we deliberately did not include the x-axis information to save space. Please note that the x-axis of those panels can be found in the corresponding panel (d) which have the same axis information. Thus, to avoid information redundancy and due to space limitation, we did not include the same x-axis three more times in the panels (a), (b), and (c).